# Antifungal Activity of Newly Formed Polymethylmethacrylate (PMMA) Modification by Zinc Oxide and Zinc Oxide–Silver Hybrid Nanoparticles

**DOI:** 10.3390/polym16243512

**Published:** 2024-12-17

**Authors:** Marek Witold Mazur, Anna Grudniak, Urszula Szałaj, Marcin Szerszeń, Jan Mizeracki, Mariusz Cierech, Elżbieta Mierzwińska-Nastalska, Jolanta Kostrzewa-Janicka

**Affiliations:** 1Department of Prosthodontics, Medical University of Warsaw, 02-097 Warsaw, Poland; marcin.szerszen@wum.edu.pl (M.S.); mariusz.cierech@wum.edu.pl (M.C.); elzbieta.mierzwinska-nastalska@wum.edu.pl (E.M.-N.); jolanta.kostrzewa-janicka@wum.edu.pl (J.K.-J.); 2Department of Bacterial Genetics, Institute of Microbiology, Faculty of Biology, University of Warsaw, 02-096 Warsaw, Poland; 33 Laboratory of Nanostructures, Institute of High Pressure Physics, Polish Academy of Sciences, 01-142 Warsaw, Poland; u.szalaj@labnano.pl (U.S.); j.mizeracki@labnano.pl (J.M.)

**Keywords:** polymethylmethacrylate (PMMA), zinc oxide nanoparticles (ZnO NPs), silver nanoparticles (Ag NPs), antifungal activity, biofilm inhibition

## Abstract

Incorporating nanoparticles into denture materials shows promise for the prevention of denture-associated fungal infections. This study investigates the antifungal properties of acrylic modified with microwave-sintered ZnO-Ag nanoparticles. ZnO-Ag nanoparticles (1% and 2.5% wt.) were synthesized via microwave solvothermal synthesis (MSS). Nanoparticles were characterized for phase purity, specific surface area (SSA), density, morphology, and elemental composition. ZnO and ZnO-Ag nanoparticles were added to acrylic material (PMMA) at concentrations of 1% and 2.5% and polymerized. Pure PMMA (control) and obtained PMMA-nanocomposites were cut into homogeneous 10 × 10 mm samples. Antifungal activity of nanoparticles and PMMA-nanocomposites against *C. albicans* was tested using minimal inhibitory concentration (MIC) determination, and biofilm formation was assessed using crystal violet staining followed by absorbance measurements. Laboratory tests confirmed phase purity and uniform, spherical particle distribution. MIC results show antifungal activity of 1% Ag nanoparticles and the PMMA-2.5% (ZnO-1% Ag) nanocomposite. PMMA-1% (ZnO-1% Ag) nanocomposite and 1% ZnO-Ag nanoparticles are efficient in preventing biofilm formation. However, ZnO nanoparticles showed antibiofilm activity, and the PMMA-ZnO nanocomposite does not protect against biofilm deposition. Incorporating hybrid ZnO-Ag nanoparticles into PMMA is a promising antibiofilm method, especially with ZnO-1% Ag nanoparticles.

## 1. Introduction

Removable prosthetic appliances are commonly used by elderly patients, whose limited manual dexterity makes it difficult to maintain adequate hygiene of their dentures. In addition, they have reduced immunity and multidrug use, which changes the oral flora. This factors promotes infections, with fungal infections being particularly significant. Infections associated with the use of prosthetic devices present a serious and real health risk for patients. Studies indicate that microorganisms colonizing dentures can cause not only local inflammation in the oral cavity, but also contribute to the development of systemic diseases, particularly respiratory infections, which can lead to severe health complications and, in extreme cases, may even be life-threatening [1,2,3,4].

The structure of acrylic resin, the material of choice for denture production, promotes fungal growth due to its porous nature, which facilitates the adhesion and subsequent colonization of the prosthetic appliance [4,5,6]. *Candida albicans* is the main etiological agent in fungal infections associated with dentures. The prosthetic device creates nonphysiological conditions in the oral cavity, weakening the natural protective barrier of the mucosa, and, combined with fungal biofilm, provides a conducive environment for infection development [4,7].

Modern methods for preventing fungal infections focus on effective denture disinfection. The most commonly used methods include soaking in disinfectant solutions, ultrasonic cleaning, and UV radiation. Moreover, new disinfection methods, such as microwave application or photodynamic therapy, show promising results in laboratory studies and may represent the future of fungal infection prevention [7,8,9,10,11]. In cases where the presence of the fungal is confirmed through mycological tests, standard treatment involves antifungal agents, mainly miconazole [12,13]. Unfortunately, the existence of strains resistant to antifungal agents and the ability of *C. albicans* to penetrate deeply into the acrylic structure makes complete eradication of the pathogen from the prosthetic material difficult [13,14]. These difficulties in treating denture-associated fungal infections increase interest in innovative methods that could improve the effectiveness of addressing this issue.

One promising research direction is to modify the material from which dentures are made by adding antimicrobial substances. Various attempts have been made to enhance acrylic materials with substances of natural and synthetic origins, such as bioactive glass, henna (*Lawsonia inermis*), chitosan, thymoquinone, *Nigella sativa* seed oil, chlorhexidine, and quaternary ammonium compounds (QACs) [13,15]. Each of these substances has antimicrobial properties, although its use is limited by adverse effects on the material’s physical properties [13]. In recent years, particular attention has been given to adding nanoparticles to acrylic materials [6,16]. Nanoparticles are structures with at least one dimension on the nanoscale, typically between 1 and 100 nm. This miniaturization increases the surface-to-volume ratio, leading to distinct chemical and physical properties compared to those of larger particles. In chemical bonding, atoms on the periphery are more reactive, resulting in unique properties for the nanomaterial [17,18].

Nanotechnology, a rapidly advancing field, has found extensive applications in industry, with its most promising potential in medicine, especially in antimicrobial and anti-cancer therapies [19,20]. In medicine, nanoparticles play a critical role in diagnostics, where they can be linked to biomarkers, facilitating imaging techniques. Additionally, their ability to selectively bind to specific cells or nucleic acids makes nanoparticles ideal drug carriers, enhancing treatment effectiveness while minimizing systemic toxicity. Nanostructures in regenerative medicine, including bone and dental tissue regeneration, have also yielded promising results. For example, studies on hydroxyapatite nanoparticles show potential to revolutionize dentistry [20,21]. Certain nanoparticles also exhibit strong antimicrobial properties. For instance, zinc oxide destabilizes microbial cell membranes and induces oxidative stress, while metals such as copper and silver disrupt enzyme production and generate reactive oxygen species (ROS). Silver and zinc oxide also demonstrate antiviral properties [6,22].

Although the modification of acrylic material with nanoparticles is not yet widely used in clinical practice, laboratory studies have yielded promising results. The addition of metal nanoparticles to the prosthetic material improves its antifungal properties [6,13,16]. Zinc oxide is an affordable and low-toxicity material that, at appropriate concentrations, does not significantly alter the physical properties of the prosthetic device [6,23]. Likewise, adding nanosilver to the acrylic material does not significantly impact mechanical properties, although the colour of the resulting material often poses a clinical acceptance issue [17,22,23]. Studies have shown that nanoparticle mixtures have more effective antimicrobial effects than single nanoparticles, opening new perspectives in combating fungal infections [19]. The literature frequently compares the effectiveness of zinc oxide and silver nanoparticles against *C. albicans*. However, to our knowledge, no studies have yet explored the modification of acrylic material using hybrid ZnO-Ag nanoparticles. This study aims to evaluate the antifungal properties of acrylic material modified with nanoparticles composed of microwave-sintered zinc oxide and silver.

## 2. Materials and Methods

### 2.1. Preparation and Characterization of Zinc Oxide and Silver Hybrid Nanoparticles

Zinc oxide (ZnO) nanoparticles and zinc oxide nanoparticles enriched with silver nanoparticles (ZnO-Ag) at concentrations of 1 and 2.5 wt.% were used for the study. The nanoparticles were produced by microwave solvothermal synthesis (MSS) using zinc acetate dihydrate (Zn(CH_3_COO)_2_∙2H_2_O, analytical pure, POCH, Gliwice, Poland) dissolved in ethylene glycol (C_2_H_4_(OH)_2_, analytical pure, CHEMPUR, Piekary Śląskie, Poland), and silver acetate (CH_3_COOAg, analytical pure, CHEMPUR, Piekary Śląskie, Poland). Microwave solvothermal synthesis of zinc oxide and metal-enriched zinc oxide were described previously [24,25,26,27,28].

The MSS process was carried out in a microwave digestion system (Ethos Up, Milestone Srl, Sorisole BG, Italy) in two stages: Stage I—heating to 220 °C at a rate of 15 °C/min (Volume 80 mL, microwave power 1600 W), and Stage II—holding at 220 °C for 35 min (Volume 80 mL, microwave power 1600 W). The obtained suspensions were centrifuged (MPW-380, MPW Med Instruments, Warsaw, Poland) to separate the solid synthesis product from the ethylene glycol. The glycol was decanted, and the precipitate was washed with deionized water. The precipitate was topped up with deionized water and intensively stirred to obtain suspensions. The centrifugation and washing process was repeated three times. The aqueous suspensions of nanoparticles (NPs) were frozen using liquid nitrogen and dried in a freeze-dryer (MARTIN CHRIST Alpha 1–4 LSCplus, Osterode am Harz, Germany). NPs of ZnO and ZnO-Ag obtained in this manner were characterized using the following methods.

X-ray powder diffraction (XRD) (X’Pert PRO, copper lamp (CuKα), Panalytical, Almelo, The Netherlands) was used to determine the phase purity of the nanoparticles obtained. The diffraction analyses were performed in 0.02° steps within the range of the 2-theta angle, from 10° to 100°, at room temperature.

Specific surface area (SSA) was tested using a surface analyser (3Flex, Micromeritics (Norcross, GA, USA)). The SSA of ZnO and ZnO-Ag nanoparticles was determined using the BET adsorption method (Brunauer–Emmett–Teller) according to ISO 9277:2010 [29]. The total pore volume of samples was estimated from the amount of adsorbed nitrogen at P/P_0_ = 0.993. The NPs samples for the density and specific surface area tests were desorbed in a degassing station (SmartVacPrep, Micromeritics, Norcross, GA, USA) for 2 h (0.05 mbar, 150 °C).

Density (DEN) was tested using a helium pycnometer (AccuPyc II 1340 V1.06, Micromeritics, Norcross, GA, USA), in accordance with ISO 12154:2014 [30]. The gas used for the measurement was helium 5.0: purity: 99.999%, moisture content 0.00006%, chemically inert gas. The measurement procedure included 100 gas purging cycles and 50 measurement cycles.

Based on the experimentally determined SSA and DEN, the Sauter mean diameter of NPs was calculated using Equation (1), with the assumption that all particles are spherical and identical [26,31].
(1)$$Sauther\;Mean\;Diameter=\frac{A}{{SSA\times 10}^{18}\times DEN\;{\times \;10}^{-21}}(nm)$$
where SMD is the Sauter Mean Diameter of nanoparticle (nm); A is a shape factor, equal to 6 for the sphere; SSA is the specific surface area (m^2^/g); and DEN is the skeletal density (g/cm^3^).

Surface morphology was investigated using a Scanning Electron Microscope (SEM) (ULTRA PLUS, ZEISS, Oberkochen, Germany) with an in-column detector (Immersion Lens detector (InLens) and Angle Selective Backscatter detector (AsB)). SEM images were obtained at magnifications of 10,000× and 150,000×. Elemental composition maps were obtained using Energy Dispersive Spectroscopy (EDS) using an X-ray spectrometer (Quantax 400, Bruker, Billerica, MA, USA).

### 2.2. Preparation of Nanocomposite Samples

Thermally polymerized acrylic material, FuturaGen (Schütz Dental, Rosbach vor der Höhe, Germany), was used for sample preparation. Using a laboratory balance (Mettler AT200, USA), the appropriate mass of nanoparticles for each group (83.3 mg for the 2.5% group and 33.3 mg for the 1% group) was measured in a silicone beaker. Subsequently, 1 g of liquid monomer was added, and the mixture was manually stirred with a spatula. The beaker was then placed in an operating ultrasonic cleaner (Elma, Singen, Germany). After mixing, 2.33 g of powder was added to achieve the proportions recommended by the manufacturer. A control sample, consisting of acrylic material without nanoparticles, was prepared in the same manner and proportions. The acrylic resin in its dough phase was placed between two smooth glass plates with 2 mm thick spacers to ensure uniform thickness. The glass plates were pressed manually, stabilized with rubber bands, and placed in a Polyclav pressure pot (Dentaurum, Ispringen, Germany) filled with water at 50 °C and 2.2 bar pressure. The obtained acrylic plates were divided into a control group (labelled as pure PMMA) and research groups with a nanoparticle content of 1% by weight or 2.5% by weight. The research groups were labelled as PMMA-1% (ZnO), PMMA-1% (ZnO-1% Ag), PMMA-1% (ZnO-2.5% Ag), PMMA-2.5% (ZnO), PMMA-2.5% (ZnO-1% Ag), and PMMA-2.5% (ZnO-2.5% Ag). Samples of dimensions 10 × 10 mm were cut out from each plate using a carborundum disc. The five most homogeneous samples were selected from each group, and their edges were smoothed with P120 alumina sandpaper and submitted for further tests.

### 2.3. Examination of Minimum Inhibitory Concentration in Nanoparticles

This study used *Candida albicans* ATCC 14053 from the collection of the Institute of Microbiology, University of Warsaw. The bacteria were cultured in Sabouraud Medium (Difco™, Pinellas Park, FL, USA) at 37 °C. The minimum inhibitory concentration (MIC) test was performed using the classical serial dilution method following the guidelines of the European Committee for Antimicrobial Susceptibility Testing (EUCAST) [32].

### 2.4. Examination of Biofilm Formation

The fungi were cultured on Sabouraud medium (Difco^TM^, Pinellas Park, FL, USA) at 37 °C. Before the experiments, overnight cultures were prepared in 10 mL of Sabouraud medium and incubated at 37 °C with shaking. The next morning, the cultures were diluted according to the McFarland scale in Sabouraud medium. The experiments were carried out on 96-well titration plates. To obtain a biofilm culture, subsequent concentrations of the tested substances were prepared, 250, 125, 62.5, 31, 15.5, 7.75, 4, 2, 1, 0.5, and 0.25 mg/L in Sabouraud medium, with 0 mg/L serving as the experimental control. The final volume was 300 µL. The prepared dilutions were inoculated with 25 µL of *C. albicans* (overnight culture diluted to 10^−4^). The biofilms were incubated statically at 37 °C for 48 h. After incubation, the medium and nonadherent fungi were carefully removed from the wells without disturbing the biofilms. The biofilm was rinsed, dried, and stained with crystal violet. The solution above the biofilms was gently aspirated with a micropipette, and the plates were rinsed with distilled water and dried at 37 °C.

Biofilm cells were stained with 0.1% crystal violet for 10 min at room temperature. Excess stain was removed by washing plates twice with distilled water, followed by drying at 60 °C. After thorough drying, 300 µL of 95% ethanol was added to each well to remove the biofilm structure. The plates were incubated for 15 min at room temperature with continuous shaking. The resulting suspensions were transferred to clean 96-well plates for absorbance measurements at 570 nm. Readings were taken using a SUNRISE microplate reader (Tecan) with Magellan software 7.3. Each experimental variant was repeated at least three times.

### 2.5. Influence of Nanocomposites on Candida albicans Solution

Overnight cultures of the reference strain *Candida albicans* 14053 in Sabouraud medium were carried out. Culture was suspended at 10^6^ cells per mL, and 2 mL of solution was added to each well of 12-well plates with pretreated denture samples. The plates were incubated for 24 h at 37 °C. The next plates were shaken on a Vortex VX-200 shaker (Labnet, Edison, NJ, USA) for 10 min. All liquid was transferred to an eppendorf tube and centrifuged at 4500× *g* for 5 min at room temperature. Cell pellet was suspended in 200 µL of fresh Sabouraud agar, and appropriate dilutions 1/10, 1/10^2^, 1/10^3^, 1/10^4^, 1/10^5^, and 1/10^6^ in PBS. 0.1 mL of each dilution was seeded in a Petri dish with a solid Sabouraud medium. The dishes were incubated for 24 h in 37 °C. The amount of colonies are presented in colony-forming units/mL (CFU/mL).

### 2.6. Biofilm Staining by Crystal Violet on Nanocomposites

Overnight cultures of reference strain *Candida albicans* 14053 in Sabouraud medium were carried out. A culture was suspended in 1:100 proportion (0.1 mL of overnight culture and 10 mL of fresh Sabouraud agar (Difco^TM^, Pinellas Park, FL, USA). An amount of 2 mL of solution was added to each well of a 12-well plate containing pretreated acrylic samples and was incubated for 24 h in 37 °C without mixing. After incubation time, samples were removed and washed 2 times in distilled water to remove nonadhered cells. An amount of 0.8% crystal violet for simple Hucker staining (CV) was prepared (0.8 g of crystal violet, 20 mL of ethanol, 0.8 g of ammonium oxalate, 80 mL of water). The stock solution was suspended (25 mL of 0.8% CV added to 175 mL of water) to obtain 0.1% CV in a working solution. An amount of 2 mL of prepared solution was added to each well of a new 12-well plate with acrylic samples. After 10 min, CV was removed, and samples were washed 2 times with distilled water and allowed to dry for 15 min at 50 °C. The pigment was dissolved by adding 600 µL of ethanol to each well and incubated at room temperature for 15 min with shaking. In total, 200 µL of the solution was transferred to a 96-well plate, and the absorbance at a wavelength of 570 nm (A570) was measured using a spectrophotometer.

## 3. Results

### 3.1. Characteristics of ZnO and Zno-Ag Nanoparticles

The XRD results (Figure 1) for the ZnO-NPs sample revealed the presence of only one crystalline phase, with all diffraction peaks assigned to the hexagonal ZnO phase (JCPDS no. 36-1451). As expected, the results for the silver-containing samples show the presence of two crystalline phases, namely the hexagonal ZnO phase (JCPDS no. 36-1451) and the cubic Ag phase (JCPDS no. 04-0783).

Figure 2 shows representative SEM images of the obtained samples. The powders of ZnO, ZnO-1% Ag, and ZnO-2.5% Ag nanoparticles consisted of homogenous particles with a spherical shape. The nanoparticles formed structures resembling a “cauliflower” shape. Elemental mapping conducted using the EDS method shows the distribution of elements within the nanopowder sample (Figure 3). For the samples with 1% and 2.5% by weight of Ag, a similar distribution of Ag ions was observed. SEM images taken using the angle-sensitive backscatter (AsB) detector (Figure 2A–C) allow for the identification of Ag NPs, visible as bright dots among the nanoparticle mixture. Several Ag aggregates are marked on the SEM image (Figure 2B,C).

Enrichment of ZnO with Ag resulted in an increase in the average size of the NPs, which correlated with a decrease in the specific surface area and a slight increase in the density of the NPs (Table 1). The increase in density is also related to the difference in theoretical density between ZnO (5.61 g/cm^3^) and Ag (10.50 g/cm^3^), as previously found [28].

### 3.2. Minimal Inhibitory Concentration of ZnO and Zno-Ag Nanoparticle Solutions Against C. albicans

The conducted MIC analyses showed high antibacterial efficacy of the nano-ZnO solution against the tested strain *C. albicans* 14053. Enrichment of nano-ZnO with silver nanoparticles at a concentration of 1% had a positive effect on the tested mixture, enhancing the killing activity of the solution against *C. albicans*. A similar effect was not observed for mixtures enriched with silver nanoparticles at a concentration of 2.5%. The activity of the solution enriched with 2.5% Ag was analogous to that of the pure ZnO solution. The results obtained are presented in the table (Table 2).

### 3.3. C. Albicans Biofilm Formed in the Presence of ZnO and Zno-Ag Nanoparticle Solutions

Analysis of the antibiofilm activity against *C. albicans* in the tested solutions showed strong inhibition of the biofilm structure formed by *C. albicans*. All the solutions tested showed antibiofilm activity even at a concentration of 5 mg/mL. The biofilm of *C. albicans*, at a concentration of 5 mg/mL compounds, was inhibited: ZnO-1% by ~29% compared to the control, ZnO-1% Ag by 32%, and ZnO-2.5% Ag by ~22%. Again, as in the case of MIC analyses, the strongest effect was demonstrated by the ZnO solution enriched with Ag1%. The results obtained for these solutions are presented in the graph below (Figure 4).

### 3.4. Influence of Acrylic-ZnO Nanocomposites Plates on Candida albicans Solution

In the results of cell viability in cultures conducted on acrylic plates enriched with the tested experimental variants, it was observed that the addition of ZnO at concentrations of 1% and 2.5% reduces the viability of *C. albicans* cells. Additional enrichment of acrylic with Ag has a beneficial effect in deepening this effect. When sowing from a 10^−6^ dilution, all experimental variants inhibited *C. albicans* by 100%. Sowing from a 10^−5^ dilution allows us to observe certain differences. The number of live cells in PMMA-1% (ZnO) compared to the PMMA control is 19%, for PMMA-1% (ZnO-1% Ag) only 2%, while for PMMA-1% (ZnO-2.5% Ag) it is 20%. All variants containing ZnO at a concentration of 2.5% (ZnO), 2.5% (ZnO-1% Ag), and 2.5% (ZnO-2.5% Ag) inhibited the growth of *C. albicans* by 100% when sown from a dilution of 10^−5^. At a 10^−4^ dilution, the highest efficacy was demonstrated by variant 2.5% (ZnO-1% Ag); in this experimental variant, there were only 2% of live *C. albicans* cells. The results obtained are presented in the form of a table of colony-forming units (Table 3) and sample photos of the number of colonies in Petri dishes (Figure 5).

### 3.5. Biofilm Staining with Crystal Violet of Acrylic Plates Containing ZnO and Zno-Ag

For further analysis, acrylic plates of dimensions 10 × 10 mm were prepared in the following variants: Pure PMMA (control) and nanocomposites: PMMA-1% (ZnO), PMMA-1% (ZnO-1% Ag), PMMA-1% (ZnO-2.5% Ag), PMMA-2.5% (ZnO), PMMA-2.5% (ZnO-1% Ag), and PMMA-2.5% (ZnO-2.5% Ag).

#### 3.5.1. Crystal Violet Biofilm Staining of Acrylic: 1% (ZnO), 1% (ZnO-1% Ag), and 1% (ZnO-2.5% Ag) Plates

The results obtained show that pure ZnO in a concentration of 1% without silver nanoparticles does not protect the acrylic surface against the deposition of *C. albicans* on it. Enrichment of the material with silver nanoparticles in concentrations of 1% and 2.5% had a positive effect on the protection of the surface tested. Biofilm was formed on the surfaces of the tested plates at a much lower level. For PMMA-1% (ZnO-1%Ag), the level of biofilm formed is 57% compared to the control on pure PMMA and 63% in the PMMA-1% (ZnO-2.5% Ag) variant. The results obtained allow us to conclude that the best antibiofilm variant against *C. allbicans* is the use of ZnO-1% enriched with 1% Ag. The results obtained are presented in the graph below and the attached example illustration of plates (Figure 6).

#### 3.5.2. Crystal Violet Biofilm Staining of Acrylic: 2.5% (ZnO), 2.5% (ZnO-1% Ag), and 2.5% (ZnO-2.5% Ag) Plates

In the further part of the analyses, the material enriched with ZnO at a concentration of 2.5% was used. The results obtained show that pure ZnO in a concentration of 2.5% without silver nanoparticles does not protect the acrylic surface from the deposition *C. albicans* on the plates. Enrichment of the material with silver nanoparticles at concentrations of 1% and 2.5% had a positive effect on the protection of the tested surface from the deposition of *C. albicans*. Biofilm was formed on the surfaces of the tested plates at a lower level, but this effect was not as strong when ZnO-1% was used as the base. For PMMA-2.5% (ZnO-1% Ag), the biofilm was at a level of 73% compared to the control on pure acrylic and 93% in the PMMA-2.5% (ZnO-2.5% Ag) variant. The obtained results allow us to conclude that the use of ZnO in a higher concentration, similarly to the use of a higher concentration of Ag, does not have a positive effect on the reduction in the biofilm formed by *C. albicans*. Of all the experimental variants used, the strongest antibiofilm effect was demonstrated by acrylic enriched with 1% (ZnO-1% Ag). During the experiments on acrylic plates, additional analyses and biological seeding were performed according to the described method (influence of nanocomposites on *Candida albicans* solution). The results obtained are presented in the form of tables and sample photos (Figure 7).

## 4. Discussion

The addition of nanoparticles to acrylic resin to change its physicochemical properties is a widely used procedure [33,34,35]. The main purpose of modifying the material for dental denture is to make it more difficult for fungal and bacterial biofilm to accumulate on it. This is to eliminate the main defect of the material, which causes inflammation of the oral mucosa and development of denture stomatitis [3]. Currently, the best known modification is the addition of silver nanoparticles, which gives very good results in reducing the growth of microorganisms in in vitro tests [36,37,38]. However, the transition to the clinical phase of research raises great doubts due to the fact that the oxidation process of silver in PMMA causes its brown discoloration [39]. This fact disqualifies the biomaterial for use in the oral cavity for esthetic reasons. This is the main reason for undertaking the research in the above article. Based on previous publications where satisfactory optical results were obtained by adding zinc oxide nanoparticles into PMMA [40,41], it was decided to synthesize a hybrid of ZnO-Ag nanoparticles, where the volume ratio of silver will not affect the esthetics of the created biomaterial (Figure 8). The ZnO-Ag hybrid synthesis has the advantage over a usual mixture of ZnO and Ag nanoparticles, as it allows for the uniform distribution of silver nanoparticles on the ZnO surface. Furthermore, co-synthesis leads to strong interactions between ZnO and Ag at the nanoscale level, resulting in a synergistic effect. Studies have shown that such nanocomposites exhibit better antibacterial and photocatalytic properties compared to ZnO and Ag mixtures [42,43,44,45,46,47]. Therefore, the use of hybrid nanoparticles with higher efficiency allows us to reduce the silver content in the composite while maintaining the antibacterial properties.

The potent microbial action of nanoparticles, in comparison to those of their micrometre counterparts, is made possible because of a very large surface area relative to the volume of the particle. This was confirmed by the study carried out by Gondal et al. [48], who determined minimal inhibitory concentration (MIC) of ZnO and ZnO NP for *Candida* spp. (10 mg/mL and 5 mg/mL, respectively). Khan et al. [49] observed that the fungicidal activity of ZnO NPs for *Candida albicans* at a concentration of 0.50 mg/mL was comparable to that of nystatine in the absence of adverse effects which may occur during antibiotic treatment. The effect of silver nanoparticles on *Candida albicans* is much stronger than that of zinc oxide, depending on the characteristics of the nanoparticles and the reference strain used. Minimal inhibitory concentration of Ag NPs is established in the range of 12–24 µg/mL [50,51]. The present study confirms the biological activity of zinc oxide nanoparticles against *Candida albicans,* the most common pathogen of denture stomatitis. The minimal inhibitory concentration value for ZnO NPs was established at 1 mg/mL. Earlier publication of the authors [52] determined the MIC for ZnO NPs for the same strain of *Candida* spp. at a similar level of 0.75 mg/mL. Differences could be due to other characteristics of ZnO NPs. The addition of nano-ZnO with silver nanoparticles at a concentration of 1% had a positive effect on the mixture tested, decreasing MIC to the 0.5 mg/mL. A similar effect was not observed for mixtures enriched with silver nanoparticles at a concentration of 2.5%. The activity of the solution enriched with 2.5% Ag was analogous to that of the pure ZnO solution. The cause of this phenomenon is unknown and requires further detailed molecular research.

The study of the nanomaterials’ effects on the liquid medium of *Candida albicans* was designed to find whether the nanoparticles may have an antifungal effect on the surrounding environment. Visible differences between individual test groups were visible at the 10^−4^ dilution of *C. albicans* colonies. The highest inhibition of *C. albicans* growth was achieved by the PMMA-2.5% (ZnO-1% Ag) nanocomposite; in this experimental variant, there were only 2% of live *C. albicans* cells compared to pure PMMA. The results obtained are consistent with the authors’ similar previous research [16], where it was observed that increasing the concentration of ZnO nanoparticles in the composites makes a better antifungal effect on the surrounding environment. To sum up this part of the experiment, it seems that the addition of Ag is most effective when silver nanoparticles are supplemented at a concentration of 1%, which is consistent with the previously obtained MIC results. The nanoparticles present in nanocomposite could possibly affect the mucosa through saliva, a constant donor of pathogenic fungi, eradicating fungi and reducing inflammation.

A procedure that prevents the formation of biofilm is a crucial component of defence against fungal infections. The prosthesis material is an excellent location for developing such consortia, which contribute to recurrent inflammations [53]. The effect of ZnO on the fungal biofilm formation was investigated using a classical method, namely crystal violet. The results obtained show that nanocomposite with pure ZnO in a concentration of 1% and 2.5% without silver nanoparticles does not protect the acrylic surface against the deposition of *C. albicans* on it, which is consistent with the publication by Cierech et al. [16], where significant antibiofilm activity was demonstrated in nanocomposites with a ZnO content above 5%. Enrichment of the material with silver nanoparticles in concentrations of 1% and 2.5% had a positive effect on the protection of the tested surface from *C. albicans* deposition. The highest antifungal activity was observed to be obtained by nanocomposites enriched with 1% Ag, whereas the level of biofilm formation decreased by 43% compared to pure acrylic. Increasing the silver content to 2.5% in the samples did not reduce the biofilm deposition. The same observation was published by Takamiya et al. [54], who discovered that incorporation of silver nanoparticles at the highest concentration (5%) was not effective in reducing CFUs, independent of particle size. This effect of Ag NPs at 5% on biofilm formation may have occurred because of changes in the polymeric surface, which have been identified as crucial factors for microorganism adhesion, their protection against shear forces, and, consequently, for biofilm growth. The same tendency was observed during the formation of biofilms on titration plates in the presence of nanoparticle solutions. Analysis of antibiofilm activity against *C. albicans* showed a strong inhibition of the biofilm structure in all tested solutions, even at a concentration of 5 mg/mL. The strongest effect was obtained for ZnO-1% Ag; biofilm formation was inhibited by ~29% compared to the control.

A limitation of the present study is the unknown cytotoxicity and mechanical properties of the obtained material. Previous studies suggest that the addition of zinc oxide to acrylic resin does not lead to a significant increase in cytotoxicity [55]. Furthermore, nanoparticles composed of ZnO doped with 1% Ag demonstrate lower cytotoxicity compared to pure ZnO nanoparticles or those with higher silver doping levels, indicating their potential safety for clinical applications [56]. The mechanical properties of the new material are also not yet known, but they are not expected to differ significantly from those of PMMA-ZnO nanocomposites. Prior to clinical application, it will also be necessary to develop environmentally friendly methods for the production and disposal of the resulting nanocomposites.

## 5. Conclusions

In this study, hybrid nanoparticles of zinc oxide and silver were successfully incorporated into PMMA resin, serving as a material for denture bases. The only visible difference between the control and the study groups after the polymerization process was a slight whitening of the material, which is fully acceptable from both a clinical and esthetic point of view. The new biomaterial gives promising results in in vitro microbiological studies using the synergism of the action of individual nanoparticles. The results obtained allow us to conclude that the best antibiofilm variant against *C. albicans* is the ZnO-1% Ag nanoparticles. The incorporation of these nanoparticles into PMMA allows us to obtain the most antifungal nanocomposite. To the best of our knowledge, this is the first successful attempt to produce PMMA resin for bases of dentures modified with nanoparticle hybrids of zinc oxide and silver. Nevertheless, a newly created nanocomposite needs to be further investigated to improve its homogeneity and to check its mechanical properties and biocompatibility prior to its possible clinical use.

## Figures and Tables

**Figure 1 polymers-16-03512-f001:**
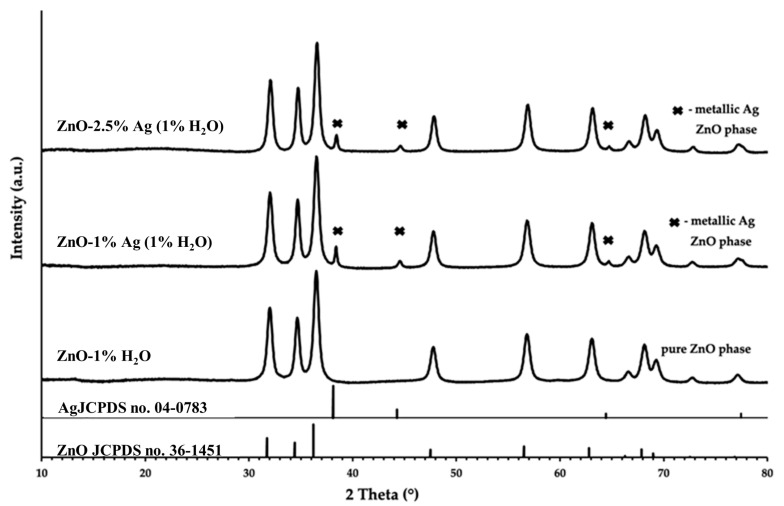
X-ray diffraction patterns of ZnO NPs and hybrid ZnO-Ag NPs.

**Figure 2 polymers-16-03512-f002:**
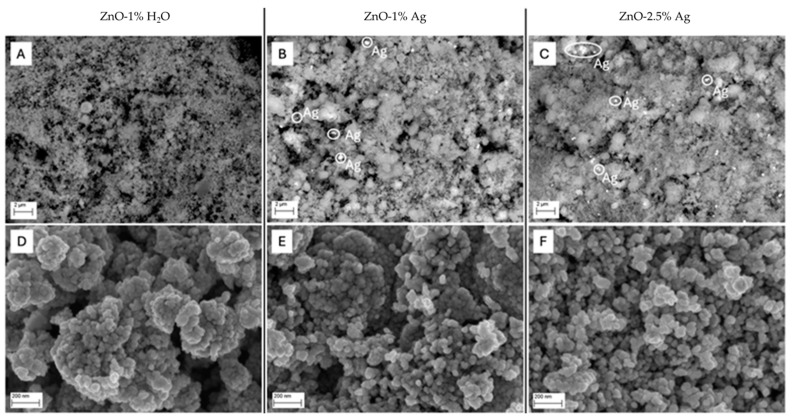
SEM images taken using the Angle-sensitive Backscatter (AsB) detector: (**A**) ZnO-1% H_2_O, (**B**) ZnO-1% Ag, and (**C**) ZnO-2.5% Ag; SEM images taken using the Immersion Lens detector (InLens): (**D**) ZnO-1% H_2_O, (**E**) ZnO-1% Ag, and (**F**) ZnO-2.5% Ag.

**Figure 3 polymers-16-03512-f003:**
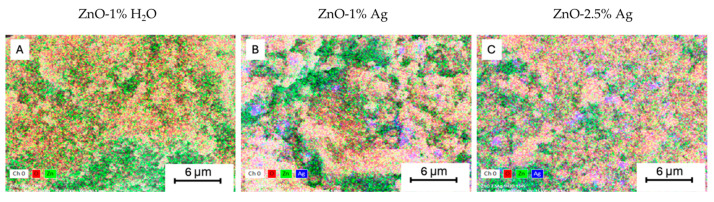
EDS images of elemental composition maps: (**A**) ZnO-1% H_2_O, (**B**) ZnO-1% Ag, and (**C**) ZnO-2.5% Ag.

**Figure 4 polymers-16-03512-f004:**
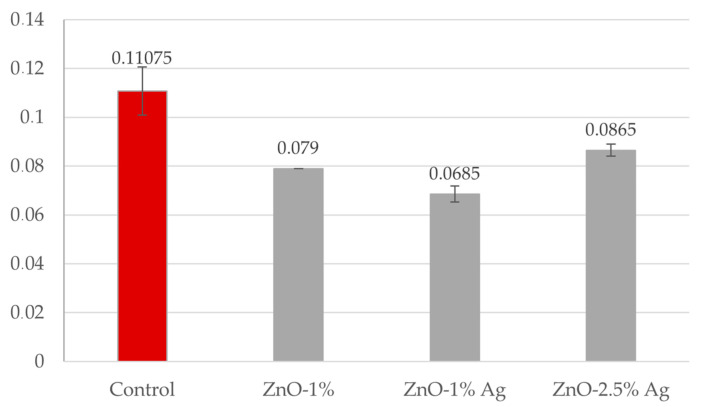
Antibiofilm activity of ZnO and ZnO-Ag nanoparticle solutions. The red control bar is a variant of *C. albicans* without the addition of nanoparticles and lines indicate standard deviation.

**Figure 5 polymers-16-03512-f005:**
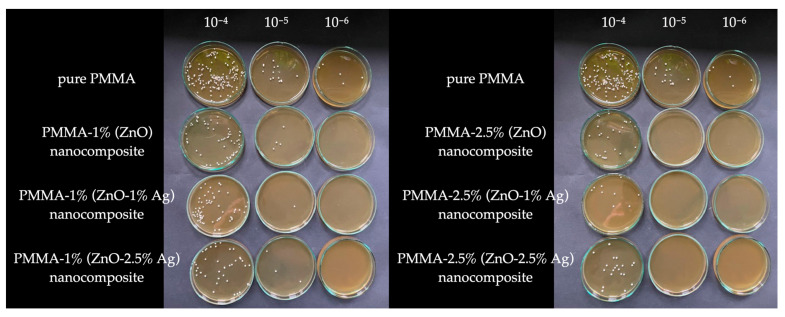
Sample photos of Petri dishes with number of *C. albicans* colonies depending on tested nanocomposite.

**Figure 6 polymers-16-03512-f006:**
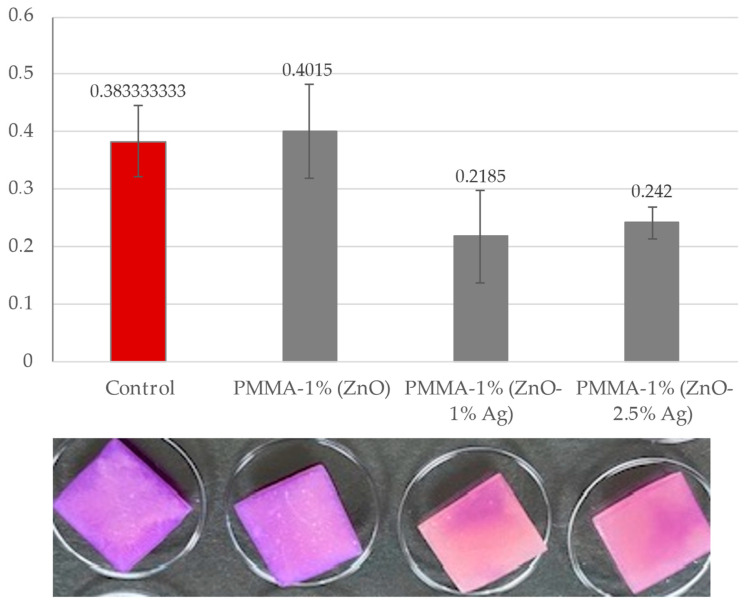
Crystal violet biofilm staining of acrylic: 1% (ZnO), 1% (ZnO-1% Ag), and 1% (ZnO-2.5% Ag) plates. The red control bar is a variant of *C. albicans* without the addition of nanoparticles and lines indicate standard deviation.

**Figure 7 polymers-16-03512-f007:**
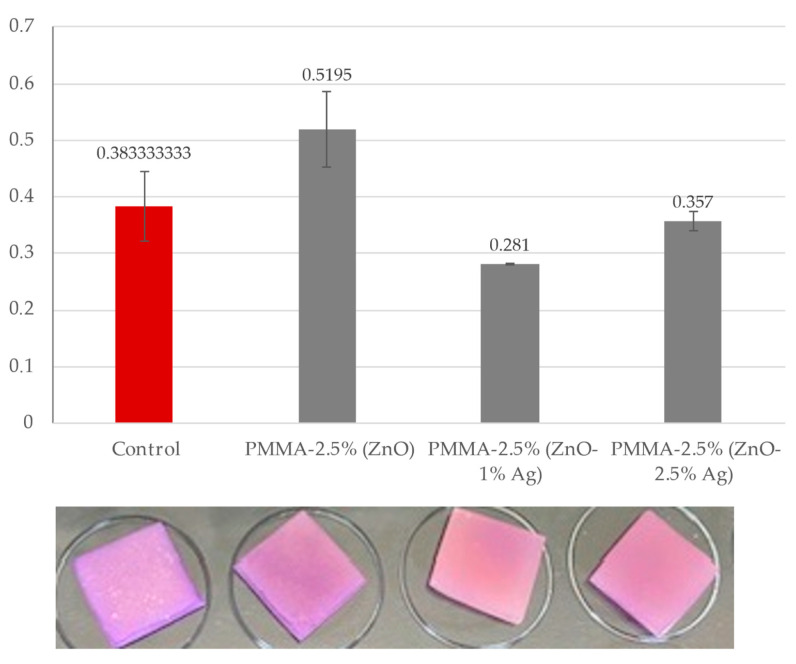
Crystal violet biofilm staining of acrylic: 2.5% (ZnO), 2.5% (ZnO-1% Ag), and 2.5% (ZnO-2.5% Ag) plates. The red control bar is a variant of *C. albicans* without the addition of nanoparticles and lines indicate standard deviation.

**Figure 8 polymers-16-03512-f008:**
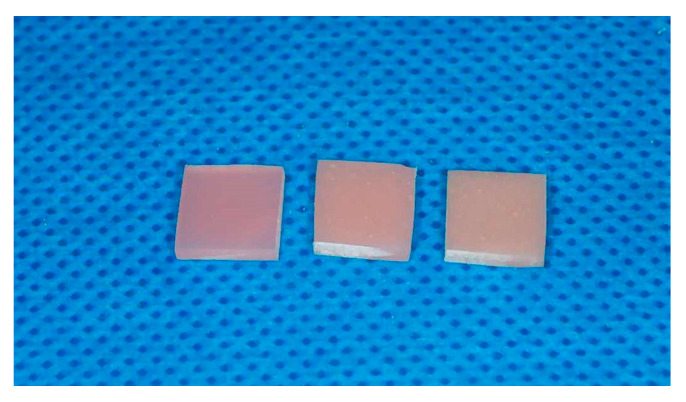
Colour differences between nanocomposites: pure PMMA (**left**), PMMA-1% (ZnO-1% Ag) (**middle**), and PMMA-2.5% (ZnO-1% Ag) (**right**).

**Table 1 polymers-16-03512-t001:** Characterization of ZnO and ZnO-Ag nanoparticles.

NPs Name	Specific Surface Area by Gas Adsorption, a_s_ ± σ (m^2^/g)	Total Pore Volume (cm^3^/g)	Skeleton Density by Gas Pycnometry, ρ_s_ ± σ (g/cm^3^)	Average Particle Size from SSA BET, d ± σ (nm)
ZnO-1%	49.0 ± 0.1	0.283	5.079 ± 0.036	24
ZnO-1% Ag	46.6 ± 0.1	0.255	5.147 ± 0.019	25
ZnO-2.5% Ag	40.6 ± 0.1	0.280	5.158 ± 0.031	29

**Table 2 polymers-16-03512-t002:** Results of minimal inhibitory concentration (MIC) analysis of ZnO and ZnO-Ag nanoparticle solutions.

Solution	MIC—mg/mL *Candida albicans* 14053
ZnO-1%	1
ZnO-1% Ag	0.5
ZnO-2.5% Ag	1

**Table 3 polymers-16-03512-t003:** The amount of colonies on tested nanocomposite plates (in CFU/mL) in variant dilutions.

	10^−4^ Dilution	10^−5^ Dilution	10^−6^ Dilution
pure PMMA	1.25 × 10^7^	1.6 × 10^7^	7 × 10^6^
PMMA-1% (ZnO) nanocomposite	6.1 × 10^6^	3 × 10^6^	0
PMMA-1% (ZnO-1% Ag) nanocomposite	7.15 × 10^6^	3 × 10^5^	0
PMMA-1% (ZnO-2.5% Ag) nanocomposite	4.4 × 10^6^	3.3 × 10^6^	0
PMMA-2.5% (ZnO) nanocomposite	1.7 × 10^6^	0	0
PMMA-2.5% (ZnO-1% Ag) nanocomposite	3 × 10^5^	0	0
PMMA-2.5% (ZnO-2.5% Ag) nanocomposite	3 × 10^6^	0	0

## Data Availability

The original data presented in the study are openly available in FigShare at https://doi.org/10.6084/m9.figshare.27924966.v2.

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
