# Peer review of "Antifungal Activity of Newly Formed Polymethylmethacrylate (PMMA) Modification by Zinc Oxide and Zinc Oxide–Silver Hybrid Nanoparticles"

_polymers, 2024, doi:10.3390/polym16243512_

Round 1

Reviewer 1 Report

Comments and Suggestions for Authors

This manuscript evaluates the antifungal efficacy of PMMA, a common denture base material, modified with zinc oxide (ZnO) and zinc oxide-silver (ZnO-Ag) nanoparticles synthesized using microwave solvothermal methods. The incorporation of 1% ZnO-Ag nanoparticles into PMMA demonstrated the best antibiofilm and antifungal activity against Candida albicans, while higher concentrations (2.5%) were less effective. Characterization confirmed the uniform distribution of nanoparticles, with minimal aesthetic compromise in the material. These findings suggest that ZnO-Ag hybrid nanoparticles can enhance the antifungal properties of PMMA, making it a promising material for reducing fungal infections in denture wearers. However, further studies on mechanical properties, biocompatibility, and optimization are necessary before clinical application.

This article is well written and research presented well. 

Some questions:

1. Why higher concentrations of Ag do not yield proportional improvements in antifungal activity?

2. What is the cytotoxicity  of ZnO-Ag nanocomposites on oral tissues.

Author Response

Thank you for your positive and encouraging feedback on our manuscript. We are delighted you found our work well-written and well-presented.

Question 1: Why higher concentrations of Ag do not yield proportional improvements in antifungal activity?

Response 1: Higher Ag concentrations do not yield proportional antifungal improvements probably due to surface property changes that affect microbial adhesion, as noted in our manuscript (line number: 431-435) and by Takamiya et al. (Ref. 52). These changes can protect against shear forces and promote biofilm formation, emphasizing the importance of maintaining optimal surface characteristics over merely increasing Ag content.

Question 2: What is the cytotoxicity  of ZnO-Ag nanocomposites on oral tissues.

Response 2: ZnO-Ag nanocomposites generally exhibit lower cytotoxicity on oral tissues compared to ZnO nanoparticles, making them promising for clinical use. However, further studies, particularly on microwave-synthesized nanocomposites, are needed to fully understand their biocompatibility. 

We have also refined the conclusions and included a paragraph on our study's limitations. Thank you again for your valuable feedback.

Reviewer 2 Report

Comments and Suggestions for Authors

Nanoworld objects have a truly amazing property to affect the mechanical properties of materials. This is why nanocomposite research is so unique and attractive. Acrylic materials are widely used in various fields, including biomedicine, and a number of works are devoted to the use of such materials in dentistry. The introduction of many nanoparticles into the polymer matrix of polymethyl methacrylate is often able to dramatically enhance the mechanical properties of the polymer matrix, and this makes such a material very interesting and promising for dentistry. The high citation rate of works in this area emphasizes the relevance and importance of this area. However, the literature lacked works that focused on the introduction of microwave-sintered ZnO-Ag nanoparticles (synthesized using microwave solvothermal synthesis) into polymethyl methacrylate. The authors fill this gap in scientific knowledge and obtain such a material, which they characterized in detail using a set of analytical methods. The authors also found antifungal activity against C. albicans and the composite also inhibits biofilm formation. The article is very well written, logical and understandable and well illustrated with high-quality figures. The experimental part is described in detail and the experiment is very carefully performed with the use of statistical methods where required. The authors use the necessary references, and the conclusions and results are in agreement with each other and with the literature. I recommend publishing after minor revision. I kindly ask the authors to discuss the safety for human health and the environmental friendliness of the obtained composites.

Author Response

Thank you for your positive and encouraging feedback on our manuscript. We are delighted that you found our work to be well-written, logical, and relevant to the field of nanocomposite research. We greatly appreciate your recognition of the importance of incorporating microwave-synthesized ZnO-Ag nanoparticles into polymethylmethacrylate and the novelty of our study in addressing this gap in the literature. Following your suggestions, we have improved the conclusions and added a paragraph on potential safety of use and environmental friendliness.